# Nanoformulations with *Leishmania braziliensis* Antigens Triggered Controlled Parasite Burden in Vaccinated Golden Hamster (*Mesocricetus auratus*) against Visceral Leishmaniasis

**DOI:** 10.3390/vaccines10111848

**Published:** 2022-10-31

**Authors:** Jennifer Ottino, Jaqueline Costa Leite, Otoni Alves Melo-Júnior, Marco Antonio Cabrera González, Tatiane Furtado de Carvalho, Giani Martins Garcia, Maurício Azevedo Batista, Patrícia Silveira, Mariana Santos Cardoso, Lilian Lacerda Bueno, Ricardo Toshio Fujiwara, Renato Lima Santos, Paulo Ricardo de Oliveira Paes, Denise Silveira-Lemos, Olindo Assis Martins-Filho, Alexsandro Sobreira Galdino, Miguel Angel Chávez-Fumagalli, Walderez Ornelas Dutra, Vanessa Carla Furtado Mosqueira, Rodolfo Cordeiro Giunchetti

**Affiliations:** 1Departamento de Parasitologia, Universidade Federal de Minas Gerais (UFMG), Belo Horizonte 31270-901, Minas Gerais, Brazil; 2Departamento de Morfologia, Universidade Federal de Minas Gerais (UFMG), Belo Horizonte 31270-901, Minas Gerais, Brazil; 3Laboratório de Desenvolvimento Galênico e Nanotecnologia, Escola de Farmácia, Universidade Federal de Ouro Preto (UFOP), Ouro Preto 35400-000, Minas Gerais, Brazil; 4Instituto Nacional de Innovación Agraria, Estación Experimental Agraria Baños del Inca, Cajamarca 06000, Peru; 5Escola de Veterinária, Universidade Federal de Minas Gerais (UFMG), Belo Horizonte 31270-901, Minas Gerais, Brazil; 6Centro de Pesquisas René Rachou/Fundação Oswaldo Cruz, Belo Horizonte 30190-002, Minas Gerais, Brazil; 7Laboratório de Biotecnologia de Microrganismos, Universidade Federal de São João Del-Rei (UFSJ), Campus Centro Oeste, Divinópolis 35501-296, Minas Gerais, Brazil; 8Computational Biology and Chemistry Research Group, Vicerrectorado de Investigación, Universidad Católica de Santa María, Urb. San José S/N, Umacollo, Arequipa 04000, Peru; 9Instituto Nacional de Ciência e Tecnologia em Doenças Tropicais—INCT-DT, Belo Horizonte 31270-901, Minas Gerais, Brazil

**Keywords:** visceral leishmaniasis, polymeric nanoparticle, vaccine, hamster, pre-clinical trial

## Abstract

Leishmaniasis is a widespread vector-borne disease in Brazil, with *Leishmania* (*Leishmania*) *infantum* as the primary etiological agent of visceral leishmaniasis (VL). Dogs are considered the main reservoir of this parasite, whose treatment in Brazil is restricted to the use of veterinary medicines, which do not promote a parasitological cure. Therefore, efficient vaccine development is the best approach to Canine Visceral Leishmaniasis (CVL) control. With this in mind, this study used hamsters (*Mesocricetus auratus*) as an experimental model in an anti-*Leishmania* preclinical vaccine trial to evaluate the safety, antigenicity, humoral response, and effects on tissue parasite load. Two novel formulations of nanoparticles made from poly(D, L-lactic) acid (PLA) polymer loading *Leishmania braziliensis* crude antigen (LB) exhibiting two different particle sizes were utilized: LBPSmG (570 nm) and LBPSmP (388 nm). The results showed that the nanoparticles were safe and harmless to hamsters and were antigenic with the induction in LBSap, LBPSmG, and LBPSmG groups of total anti-*Leishmania* IgG antibodies 30 days after challenge, which persists 200 days in LBSap and LBPSmP. At the same time, a less pronounced hepatosplenomegaly in LBSap, LBPSmG, and LBPSmP was found when compared to control groups, as well as a less pronounced inflammatory infiltrate and granuloma formation in the spleen. Furthermore, significant reductions of 84%, 81%, and 90% were observed in spleen parasite burden accessed by *q*PCR in the LBSap, LBPSmG, and LBPSmP groups, respectively. In this way, LBSap, LBPSmG, and LBPSmP formulations showed better results in vaccinated and *L. infantum*-challenged animals in further reducing parasitic load in the spleen and attenuating lesions in liver and splenic tissues. This results in safe, harmless nanoformulation vaccines with significant immunogenic and infection control potential. In addition, animals vaccinated with LBPSmP had an overall reduction in parasite burden in the spleen, indicating that a smaller nanoparticle could be more efficient in targeting antigen-presenting cells.

## 1. Introduction

Leishmaniasis is a complex of parasitic diseases transmitted during female sand flies’ blood meal by *Phlebotomus* genus in the Old World and *Lutzomyia* in the New World. Seven countries account for 90% of the cases distributed worldwide: Brazil, Ethiopia, India, Kenya, Somalia, South Sudan, and Sudan. The disease, which can be caused by about twenty different species of *Leishmania* parasites, has three main clinical manifestations: cutaneous, mucosal, and visceral [1]. *L. infantum* is the main etiological agent involved in visceral leishmaniasis (VL) in Latin America, although some studies report dogs with VL clinical symptoms that were found positive for *L. amazonensis* or *L. braziliensis* infection [2,3,4,5,6,7,8].

In Brazil and many Europe countries—primarily in the Mediterranean basin—dogs are considered the main reservoir of *L. infantum*, and a large percentage of them are asymptomatic, which contributes to the spread of the disease [9]. In this sense, canine visceral leishmaniasis (CVL) represents a serious public health problem, whereas control of human cases is closely linked to the breakdown of the disease cycle in dogs [10]. Control measures, such as the use of impregnated insecticide collars, raising people’s awareness, and new vaccine technologies, are rational approaches that could be implemented to control CVL. 

Nanotechnology-based formulations have emerged as a safe and powerful alternative to be used in the control of various diseases, promoting more assertive drug or vaccinal antigen deliveries. This approach is able to target cells and, for vaccinal antigens, trigger the immune system, thereby improving the immunogenicity without adjuvants in the formulation [1,11,12,13]. In this sense, polymeric nanoparticles are a good alternative to be used in VL control as carriers of *Leishmania* antigens. Depending on the polymer, they exhibit low toxicity and good cost benefits. Furthermore, PLA-based nanocarriers are fully biodegradable and stable in biological environments, delaying antigen release before they reach the target cells [1,14].

Likewise, the choice of the appropriate experimental model is relevant in immunobiological research, with the golden hamster (*Mesocricetus auratus*) properly mimicking the VL course in its vertebrate hosts, such as human and dogs, unlike BALB/c mice [15,16].

The present study evaluated the association between two polymeric nanoformulations with *L. braziliensis* antigens as a potential immunobiological component against CVL using golden hamsters as an experimental model. Clinical, hematological, immunological, parasitological, histological and molecular parameters were evaluated to better assess the efficacy of the vaccine candidates in the animal model. 

## 2. Materials and Methods

### 2.1. Ethical Statement

This project was approved by the Ethics Committee on the Use of Animals of the Federal University of Minas Gerais (CEUA-UFMG) by protocol 385/2013.

### 2.2. Studied Animals and Experimental Groups

Seventy-two healthy male golden hamsters (*Mesocricetus auratus*) between 6 and 8 weeks of age were divided into eight groups with eight animals each. The groups were established as follows: infected control group (C), saponin inoculated group (Sap), *Leishmania braziliensis* antigen vaccinated group (LB), *L. braziliensis* antigen plus saponin vaccinated group (LBSap), empty submicrometric large particle vaccinated group (PSmG), submicrometric large particle plus *Leishmania braziliensis* crude antigen vaccinated group (LBPSmG), empty submicrometric small particle vaccinated group (PSmP), and submicrometric small particle plus *Leishmania braziliensis* crude antigen vaccinated group (LBPSmP). Eight animals were used as a preimmune group.

### 2.3. Production of Vaccine Formulations

To prepare the *Leishmania braziliensis* antigen, a reference strain (MHOM/BR/75/M2904) was grown in Minimum Essential Medium (α–MEM-Gibco BRL, Waltham, MA, USA) supplemented with 10% bovine fetal serum and 2 mM penicillin under biochemical oxygen demand (BOD) at 23 °C for a maximum of 10 passages. On the fifth day, 10^8^ stationary promastigotes were sonicated, and the antigen was aliquoted and stored at −80 °C [17]. Previous studies from our group had already demonstrated that *L. braziliensis* crude antigen, combined or not to saponin, are safe and innocuous in in vitro, pre-clinic and clinic models and are promising vaccines against CVL [17,18,19].

Saponin was used as the adjuvant (Sigma-Aldrich^®^, St. Louis, MI, USA) at a 100 μg/dose concentration diluted in phosphate saline buffer (PBS) at the inoculum time. 

The nanospheres were prepared at Laboratório de Desenvolvimento Galênico e Nanotecnológico of the Escola de Farmácia from Universidade Federal de Ouro Preto by using the interfacial deposition method of a preformed polymer, as described by Fessi et al. [20].

Empty nanospheres were prepared by associating a poly(D, L-lactic) acid polymer (PLA) solution in an organic solvent with chitosan dissolved in 0.05 M acetic acid at concentrations of 50 mg/mL (large submicrometric particle) and 5 mg/mL (small submicrometric particle). Antigen-loaded nanoparticles (NP) were obtained using these same solutions by adding 1 mL of the *L*. *braziliensis* crude antigen suspension at a concentration of 1 mg/mL. The nanospheres were characterized by size, polydispersity index, and zeta potential, as described by Garcia et al. [21]. Large submicrometer particles (PSmG—empty particle) were 613 nm in sizes and small submicrometer particles (PSmP—empty particle) were 35 nm in size. The antigen encapsulated NP showed a mean size of 570 nm for large submicrometer particles (LBPSmG) and 388 nm for small submicrometer particles (LBPSmP).

Each vaccine formulation was diluted with 0.9% sterile saline buffer during its preparation, and the final volume was adjusted for 100 µL/dose.

### 2.4. Experimental Protocol

The animals enrolled in the study (*n* = 72 hamster; *n* = 8/group) were kept in ventilated racks with food and water ad libitum throughout the study, with intermittent light/dark cycles every 12 h, submitted to fully blinded preclinical interventions. The hamsters received a dose of the vaccine subcutaneously according to their respective groups, in intervals of 21 days up to three doses. The safety and toxicity evaluations of the nanoformulations were performed by observing the inoculum site (e.g., analysis of presence of nodules, papules, or ulcerative lesions), by evaluating animal’s behavior (hostility or apathy), and also by catching sight of signs such as local pain, alopecia, and bristling. Thirty days after beginning vaccine protocol (T30), the first blood sample of the animals was taken and, after 50 days (end of the vaccine protocol), the animals were challenged with 10^7^ *L. infantum* stationary promastigotes (strain C46) resuspended in saline in a 500 µL final volume by intraperitoneal route [22]. *L*. *infantum* culture was grown under the same conditions described by Giunchetti et al. [17]. 

The hamsters were surveyed throughout the experimental period for the presence of lesions at the inoculum and challenge sites; clinical signs such as cachexia, alopecia, organs enlargement; behavioral changes; and weight loss. Two hundred days post challenge (T200), a new blood sample was taken, and the animals were euthanized by sodium pentobarbital overdose (150 mg/Kg), injected intraperitoneally.

### 2.5. Assessment of Total Anti-Leishmania IgG Antibodies

Total IgG antibodies were measured through an in house ELISA assay according the protocol described by Moreira et al. 2012 [23]. The 96-well plates were sensitized with 1 µg/well of soluble *L. infantum* strain C46 antigen. Unspecific ligations were avoided by adding 200 μL/well of a PBS solution containing 3% BSA. The hamster’s serum was added using a 1:100 proportion and the secondary anti-hamster IgG antibody was diluted 1:5000 times, both in PBS Tween 20 in a 0.05% concentration. The reaction was revealed with 0.03% hydrogen peroxide and 0.022 µg of ortho-phenylenediamine (OPD). An ELX800 (Biotek Instruments VT, Winooski, VT, USA) spectrophotometer was used to measure the wells’ luminescence in a 490 nm wavelength. The cut-off was established by the average of the optical density (OD) of eight non-infected hamsters (pre-immune group related to randomized samples in hamster of all groups before the immunization protocol) plus two standard deviations measured for these samples. 

### 2.6. Histology Analysis

Fragments of the liver and spleen from 64 animals (except for the pre-immune group) were analyzed by histopathology stained with Hematoxylin and Eosin. The liver was analyzed for inflammatory lymphohistiocytic infiltrate, lymphohistiocytic and neutrophilic inflammatory infiltrate, granuloma, hydropic degeneration, and cholestasis. The spleen was evaluated for the presence of inflammatory lymphohistiocytic infiltrate, lymphohistiocytic and neutrophilic inflammatory infiltrate for red and white pulps, lymphoid hyperplasia, granuloma, amyloidosis, amastigotes, and perisplenitis. 

### 2.7. Parasite Burden Evaluation

#### 2.7.1. Limiting Dilution Assay

Vaccine efficacy was assessed by parasitological assay using limiting dilution in the liver and spleen samples. The organs from all the animals of each group (except for the pre-immune group) were removed in sterile condition, weighed (for total and fragment organ weight), macerated with 2 mL of α–MEM (minimum essential medium) in a tissue macerator (Pyrex^®^, Burlington, MA, USA), and centrifugated at 1540× *g* at 4 °C for 10 min. The liver and spleen homogenates were plated in duplicates in a 96-well microtitre plates in serial 5-fold successive dilutions. All the plates were maintained for 7 days in a biochemical oxygen demand incubator at 23 °C. The average parasite count in the well duplicate, where it was last visualized, was multiplied by the organ’s weight and the dilution factor corresponding to the well. The value obtained corresponded to the parasite burden in the respective organ fragment. It was then possible to estimate the parasite burden in the organs [24,25].

#### 2.7.2. Quantitative PCR (*q*PCR)

The parasite gDNA extraction from the spleen was performed by using a NucleoSpin^®^ Tissue extraction kit—Genomic DNA from tissue protocol (Macherey-Nagel, Duren, Germany) according to the manufacturer’s instructions. DNA was resuspended in 50 μL of elution buffer (kit provided) and quantified using Nanodrop N2000 equipment (Thermo Scientific, Waltham, MA, USA). The *q*PCR reactions were performed using 5 μL of the 2× Power Syber^®^-Green ABI reagent (Thermo Fisher, USA), 0.5 μL of each primer (2 μM), (kDNA.*Leish*.F: 5′-CGTGGGGGAGGGGCGTTCT-3′ and kDNA.*Leish*.R: 5′-CCGAAGCAGCCGCCCCTATT-3′) with a 135 pb amplicon size [26], and 4 μL of DNA (5 ng/μL) in a final volume of 10 μL per well. The β-actin constitutive hamster gene was selected to normalize the reaction using primers that amplify a 120 bp fragment (Forward: 5′-TCGTACGTGGGTGACGAGGC-3′; reverse: 5′-GTAGAAGGTGTGGTGCCAGA-3′). The amplification process took place following the cycles: initial denaturation at 95 °C for 10 min, followed by 40 cycles of denaturation at 95°C for 15 s, and a melt curve stage at 60 °C for 1 min. 

Finally, the *q*PCR analysis was performed using 7500 Software v2.0.1 (Thermo Fisher, USA), and the results were plotted using the standard curve method (parasite burden was inferred through the difference between the amount of b-actin kDNA and the amount of the hamster’s sample kDNA). They were assembled from the serial dilution of DNA extracted from 10^8^ *Leishmania* sp. promastigotes and 10^6^ hamster—*Mesocricetus auratus*—splenocytes. 

### 2.8. Statistical Analyses

Statistical analyses were conducted using the Prism 5.0 software package (Prism Software, Irvine, CA, USA). Data normality was assessed using the Kolmogorov–Smirnoff test. One-way analysis of variance (ANOVA) and Tukey post-test were used to investigate differences between groups. Considering the nonparametric nature of real time PCR data, the Kruskal–Wallis test was used, followed by Dunn’s test. Statistical significance was considered at a *p* value < 0.05. 

## 3. Results

### 3.1. Immunobiological Safety and Harmless Analysis

The animals were observed after each vaccine dose for possible post-vaccination reactions, and the vaccine was shown to be safe and innocuous. Subsequent to animal challenge, the control groups (C, Sap, LB, PSmG, and PSmP) displayed more evident dermatological changes (e.g., alopecia) as compared to animals vaccinated with formulations that contained the *Leishmania* antigen (LBSap, LBPSmG, and LBPSmP). Notably, the controls groups (C, Sap, LB, PSmG, and PSmP) exhibited enlarged livers and spleens after euthanasia.

### 3.2. Immunogenic Potential of Nanoformulations

The LBPSmG and LBPSmG groups showed increased total anti-*Leishmania* l IgG levels 30 days after two vaccine boosters as compared to the other groups (C, Sap, LB, PSmG, and PSmP—*p* < 0.05) (Figure 1A). Furthermore, the LBSap group still presented high levels of total IgG antibodies 200 days after experimental challenge with *L. infantum as* compared to the LB, PSmG, and PSmP groups (*p* < 0.05) (Figure 1B), and the LBPSmP group showed elevated levels of IgG antibodies as compared to the PSmP group (*p* < 0.05) (Figure 1B).

### 3.3. Evaluation of Changes in Parasite Target Tissues

The lesions on the liver of *L. infantum*-infected animals were accessed by frequency histopathological evaluation, and the LBPSmP group showed a decrease in cholestasis as compared with the LB group, as shown in the bar graphs in Figure 2A–D/ and images in Figure 2A′–D′.

The descriptive histopathological analysis of the spleen revealed a significant reduction in granuloma progression in LBSap and PSmG as compared to animals vaccinated only with the LB antigen. Additionally, perisplenitis was reduced in the Sap, LBSap, and PSmP groups compared to the C group. The frequency in which histological changes occur by group is presented in bar graphs (Figure 3A–G) and shown in representative images in Figure 3A′–G′. 

### 3.4. Efficacy Evaluation Based on Parasite Load

#### 3.4.1. Limiting Dilution Assay

In addition to a tendency toward reduced parasitism in the liver (Figure 4A) in the LBSap and LBPSmP groups, the lowest parasitic load was observed in the spleen in the vaccine groups (LBSap, LBPSmG, and LBPSmP) as compared to the control groups by limiting dilution assay (Figure 4B).

#### 3.4.2. Quantitative Molecular Analysis (*q*PCR) 

Data obtained in the *q*PCR analysis showed that the LBPSmP, LBSap, and LBPSmG groups have the lowest parasite burden/10^6^ hamster splenocytes compared to the other groups (Figure 5). Notably, the percentage of parasite load reduction in the LBSap, LBPSmG, and LBPSmP groups were 90%, 88%, and 94%, respectively, when compared to the SAP group; 88%, 86%, and 93%, respectively, when compared to the LB group; and 84%, 81%, and 90%, respectively, when compared to the C (Control) group.

## 4. Discussion

The development of vaccines for pathogen control is a hallmark in modern medicine. Nanovaccines have been studied due to their efficient antigen delivery systems and ability to reach a specific target [27,28,29]. In addition, this technology has the added advantage of having no potential side effects as compared to the adjuvants used in the development of immunobiological candidates with less local and systemic side effects. 

This work combines two novel nanoparticles associated with the *L. braziliensis* antigen as an alternative for visceral leishmaniasis control. Previous works have already demonstrated the cross protection of *L. braziliensis* crude antigens against *L. infantum* infection, as well as the genetic homology between these parasite species [17,18,19]. Giunchetti et al. (2007) [17] demonstrated a significant increase in nitric oxide (NO) levels in dogs vaccinated with LBSap in an in vitro cell proliferation assay stimulated with soluble *L. chagasi* antigen. This study demonstrated a protective response triggered by *L. braziliensis* plus saponin vaccine against visceral leishmaniasis. Furthermore, hamsters were chosen as experimental models because the evolution of their disease is similar to that in dogs and humans [30,31,32].

In this sense, clinical and laboratory studies have shown that the combination between nanoparticles and *L. braziliensis* antigen are safe and harmless after the three-dose vaccine protocol as no clinical, behavioral, or significant changes in weight were observed in hamsters. LBSap, an extensively studied vaccine, also presented no toxicity and deemed to be safe, as previously demonstrated. [17,18,19,33,34,35,36,37]. After *L. infantum* experimental challenge, and before the animals were euthanized, some dermatological disorders, such as alopecia, were observed in the C, Sap, LB, PSmG, and PSmP groups, which are considered a common clinical sign of VL [38,39].

Increased levels of total anti-*Leishmania* IgG antibodies were observed in this work 30 days after *L. infantum* challenge, which persisted 200 days after challenge in animals from the LBSap and LBPSmP groups. 

These data demonstrate that the vaccines triggered a strong antigenicity profile, which was described in studies conducted in hamsters [40], mice [35], and dogs [17,33,34,37]. Nanoparticle size may also be directly related to immunogenic vaccine potential, whose physical-chemical characteristics have been extensively studied [27,41].

The LB and PSmG groups showed an increase in cholestasis and inflammatory cell infiltration intensity, indicating hepatic tissue damage. The presence of damaged hepatic ducts had already been observed in mice and preceded any other inflammatory processes in hepatic tissue, such as fibrosis and steatosis. This may lead to Küpffer cell activation with IL-6 production, along with fibrin and collagen deposition in the bile ducts, causing blockage to the liver [42]. Moreover, inflammatory cells move to the parasitized site, orchestrated by immune mechanisms, in order to enclose the infection, culminating in granuloma formation both in hamsters and dogs [38,39,43,44]. However, a reduction in these changes was clearly observed in the LBPSmP group. Hepatosplenomegaly was notably absent or less prominent in the LBSap, LBPSmG, and LBPSmP groups [15,39,45,46]. The presence of *Leishmania* amastigotes in parasitized tissues may induce changes in their structure and physiology [47].

In healthy individuals, the spleen is the lymphoid organ foremost related to circulating pathogen defense, and its structure may be severely affected during chronic VL disease [48]. The elimination of amastigotes in this organ is inefficient in hamsters [49,50], and some studies suggest that a disruption in spleen compartments may occur with disorders of the red and white pulp [48,51,52], which were observed in the control groups of the present study. However, the granuloma formation was slightly diminished, especially in LBSap and LBPSmP groups, and could be associated with the activation of the immune system and reduction in parasite load [44,49].

In fact, the parasite burden quantification assessed by limiting dilution and *q*PCR assays showed a significant reduction in *Leishmania* parasite burden in the LBSap, LBPSmG, and LBPSmP groups, demonstrating the efficiency in controlling parasite spread in hamsters. Similarly, in another LBSap preclinical vaccine trial using BALB/c mice, a 42% decrease in *Leishmania* parasites was observed in the spleen of vaccinated animals [35] as compared to the 84% reduction in hamsters in this study. Similarly, Roatt et al. [52] also reported a reduction in parasite burden in bone marrow, skin, and sand flies in a study using an *L. braziliensis* antigen (associated to MPL adjuvant (LBMPL)) as a vaccine in an immunotherapeutic study in naturally *L. infantum*-infected dogs. Importantly, the groups that presented the lowest amount of enlargement of the liver and spleen—LBSap, LBPSmG, and LBPSmP—also showed the lowest parasitic load, similar to what Lima et al. demonstrated in asymptomatic animals [53].

The LBSap, LBPSmG, and LBPSmP formulations demonstrated better results in vaccinated and *L. infantum* challenged animals in reducing the parasite load, making it a promising vaccine candidate. Significantly, a hallmark profile in the LBPSmP formulation includes biomarkers related to restricted parasite load and low amount of histological damage between the vaccinated groups.

## 5. Conclusions

The vaccine formulations proposed in this study (LBSap, LBPSmG, and LBPSmP) were innocuous, safe, and antigenic in hamsters. Moreover, they proved efficient in triggering total anti-*Leishmania* IgG antibodies and in attenuating tissue damage in the liver and spleen, especially LBSap and LBPSmP. Although immunophenotyping assays were not performed, the parameters evaluated in the vaccinated animals suggest the occurrence of a protective cellular profile in this pre-clinical assay. In addition, a significant parasite load reduction was observed in the spleen of hamsters vaccinated with LBPSmP indicating that a smaller nanoparticle could be more efficient in delivering the antigen to the target cells. These results should stimulate additional studies, especially regarding the nature of the immune response triggered by the nanoparticles and the mechanisms involved. The use of nanotechnology associated with *Leishmania* antigens could be an important strategy for visceral leishmaniasis control measures.

## Figures and Tables

**Figure 1 vaccines-10-01848-f001:**
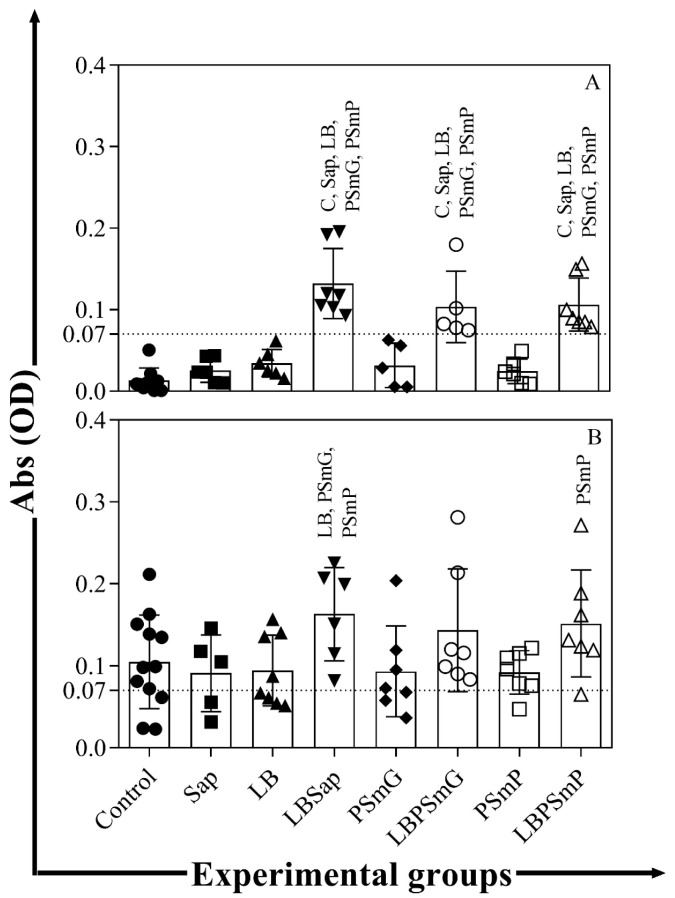
**Evaluation of anti-Leishmania infantum humoral immune response in hamsters from the different experimental groups**. Serum samples were obtained from the animals 30 days after the third dose of the vaccines (**A**) and 200 days after the experimental challenge (**B**). The graphs represent the mean values and standard deviations of total IgG optical density measured by the ELISA assay for each group. Statistical differences (*p* < 0.05, Mann–Whitney) between groups are indicated by acronyms in the correlated group(s). LEGEND: C—Control (full circle); Sap—Saponin (full square); LB—*L. braziliensis* antigen (full triangle); LBSap—*L. braziliensis* antigen plus saponin (full inverted triangle); PSmG—Empty submicrometric large particle (full diamond); LBPSmG—Submicrometric large particle plus *Leishmania braziliensis* antigen (empty circle); PSmP—Empty submicrometric small particle (empty square); LBPSmP—Submicrometric small particle plus *Leishmania braziliensis* antigen (empty triangle).

**Figure 2 vaccines-10-01848-f002:**
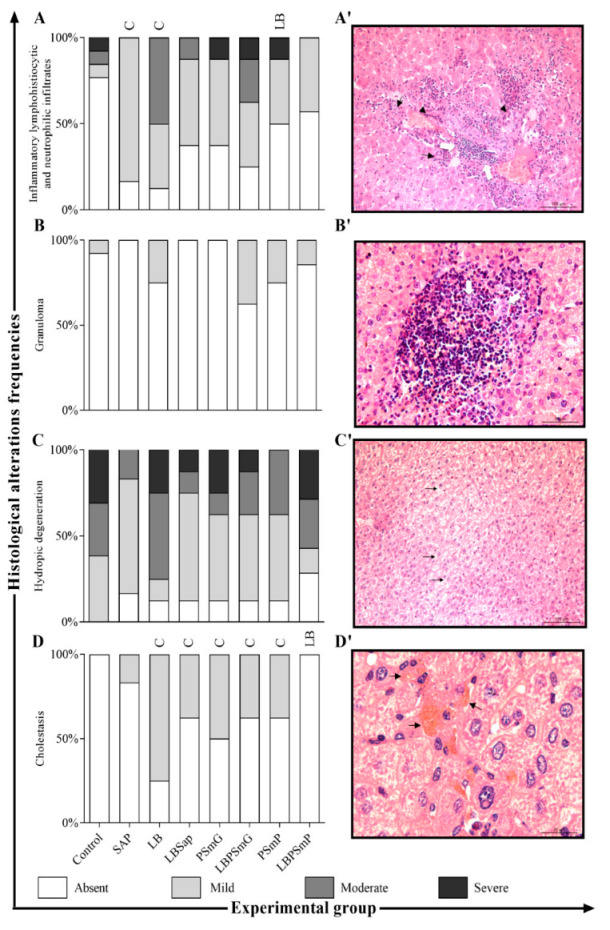
Histological changes in the liver after experimental *L. infantum* challenge in slides stained by Hematoxylin and Eosin. On the left: The *x* axis represents the frequency of each histological change (**A**–**D**) according to the immunization group (Control—C, Sap, LB, LBSap, PSmG, LBPSmG, PSmP, and LBPSmP) shown in the *y* axis. On the right: The photomicrographs refer to the main histological changes found in hamster livers after experimental challenge. (**A′**)—Inflammatory lymphohistiocytic and neutrophilic infiltrates: hepatitis characterized by inflammatory lymphohistiocytoplasmic (white arrows) and neutrophilic (head arrows) infiltrates with rare multinucleated giant cells (black arrows) (20× magnification); (**B′**)—Granuloma: lymphohysto-plasma-cell and neutrophilic hepatitis with some epithelioid macrophages (white arrows), characterizing hepatic micro-granulomas (40× magnification); (**C′**)—Hydropic degeneration: multifocal, discrete hydropic degeneration evidenced by the presence of intracytoplasmic vacuoles in hepatocytes (black arrow) (20× magnification); (**D′**)—Cholestasis: accumulation of intracytoplasmic pigment in hepatocytes (head arrow), compatible with bile pigment, multifocal, discrete, associated with moderate multifocal random lymphohistiocytic inflammatory infiltrate and hydropic, multifocal, discrete degeneration (100× magnification). Statistical differences (*p* < 0.05, Mann–Whitney) between groups are indicated by acronyms in the correlated group(s).

**Figure 3 vaccines-10-01848-f003:**
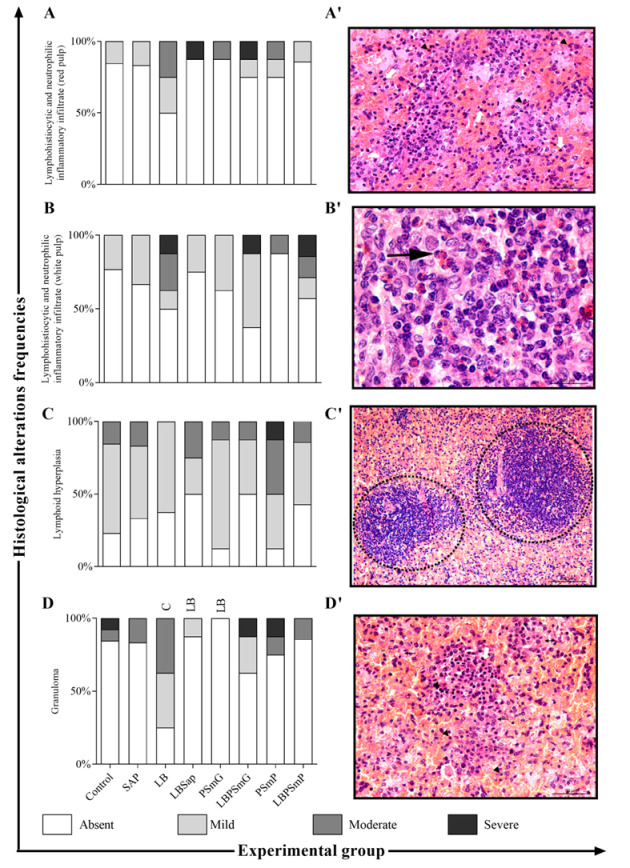
Histological changes in the spleen after *L. infantum* experimental challenge in slides stained by Hematoxylin and Eosin. On the left: The *x* axis represents the frequency of each histological change (**A**–**G**) according to the immunization group (Control—C, Sap, LB, LBSap, PSmG, LBPSmG, PSmP, and LBPSmP) shown in the *y* axis. On the right: The photomicrographs refer to the main histological changes found in hamster spleens after experimental challenge. (**A′**)—Lymphohistiocytic and neutrophilic inflammatory infiltrate (red pulp) (head arrows): inflammatory histiocytic inflammatory with some neutrophils (white arrows), multifocal, moderate (40× magnification); (**B′**)—Lymphohistiocytic and neutrophilic inflammatory infiltrate (white pulp): moderate multifocal inflammatory infiltrate (black arrow) (100× magnification); (**C′**)—Lymphoid hyperplasia: discrete multifocal lymphoid hyperplasia associated with moderate diffuse hyperemia (20× magnification) (circled areas); (**D′**)—Granuloma: inflammatory histiocytic infiltrate, with some lymphocytes (head arrows) and plasmacytes (white arrow) and rare multinucleated giant cells, forming microgranulomas, multifocal to coalescent, intense, associated with the deposition of eosinophilic (black arrows), amorphous and acellular material in red pulp, compatible with amyloidosis, diffuse and moderate (40× magnification); (**E′**)—Amyloidosis: deposition of mild eosinophilic, amorphous and acellular material in the red pulp (black arrows), compatible with intense diffuse amyloidosis (100× magnification); (**F′**)—Amastigotes: microgranulomas containing intracytoplasmic *Leishmania infantum* amastigotes (black arrow) in histiocytes (increase of 100×); (**G′**)- Perisplenitis: discrete multifocal lymphohistiocytic perisplenitis (lymphocytes—head arrows; and histiocytes—white arrows) (40× magnification). Statistical differences (*p* < 0.05, Mann–Whitney) between groups are indicated by acronyms in the correlated group(s).

**Figure 4 vaccines-10-01848-f004:**
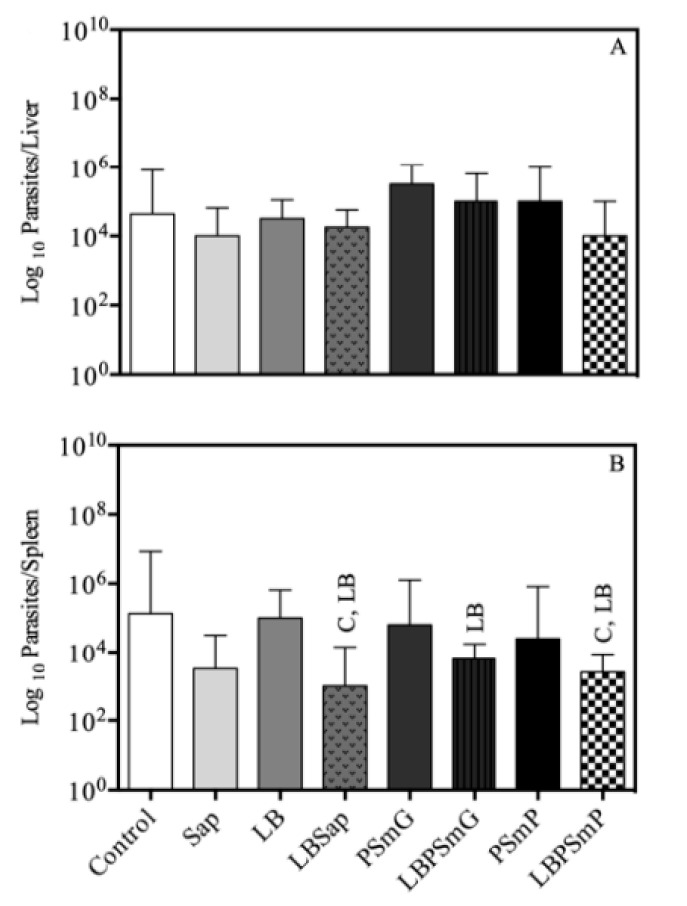
**Parasite burden in liver (A) and spleen (B) after experimental challenge by limiting dilution assay.** The *x* axis represents the experimental groups and the *y* axis represents the mean and standard deviation in log units of parasite burden in liver and spleen. Statistical differences (*p* < 0.05, Mann–Whitney) between groups are indicated by acronyms in the correlated group(s). LEGEND: C—Control (white bar); Sap—Saponin (light gray bar); LB—*L. braziliensis* antigen (medium gray bar); LBSap—*L. braziliensis* antigen plus saponin (dot medium gray bar); PSmG—Empty submicrometric large particle (dark gray bar); LBPSmG—Submicrometric large particle plus *Leishmania braziliensis* antigen (vertical line dark gray bar); PSmP—Empty submicrometric small particle (black bar); LBPSmP—Submicrometric small particle plus *Leishmania braziliensis* antigen (checkered bar).

**Figure 5 vaccines-10-01848-f005:**
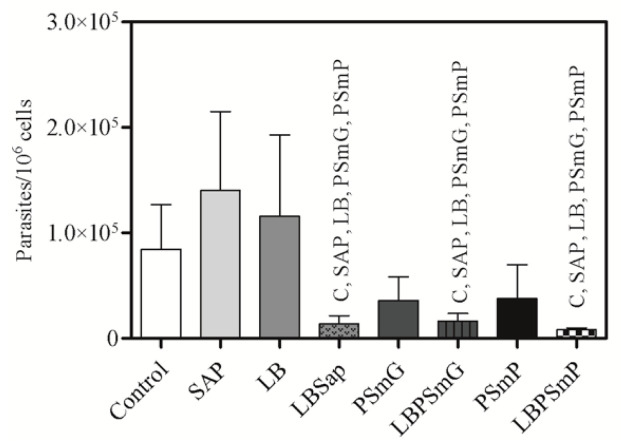
**Parasite load in spleen after experimental challenge obtained from *q*PCR reaction.** The results represent the log scale of parasite burden quantification in the spleen. The *x* axis represents the experimental groups and the *y* axis represents the mean and standard deviation of parasite quantification in 10^6^ splenocytes. Statistical differences (*p* < 0.05, Mann–Whitney) between groups were indicated by acronyms in the correlated group(s). LEGEND: C—Control (white bar); Sap—Saponin (light gray bar); LB—*L. braziliensis* antigen (medium gray bar); LBSap—*L. braziliensis* antigen plus saponin (dot medium gray bar); PSmG—Empty submicrometric large particle (dark gray bar); LBPSmG—Submicrometric large particle plus *Leishmania braziliensis* antigen (vertical line dark gray bar); PSmP—Empty submicrometric small particle (black bar); LBPSmP—Submicrometric small particle plus *Leishmania braziliensis* antigen (checkered bar).

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
