# Peer review of "Nanoformulations with Leishmania braziliensis Antigens Triggered Controlled Parasite Burden in Vaccinated Golden Hamster (Mesocricetus auratus) against Visceral Leishmaniasis"

_vaccines, 2022, doi:10.3390/vaccines10111848_

Round 1

Reviewer 1 Report

 I checked carefully the title "Nanoformulations with Leishmania braziliensis antigens triggered controlled parasite burden in vaccinated golden hamster (Mesocricetus auratus) against visceral leishmaniasis" 

Authors are presenting the evidence that vaccine based in L. braziliensis antigen presented using microparticles are more efficent then vaccine presented using the saponine adjuvant. 

It is a manuscript that could be of interest of the reader of Vaccine journal, despite of it it needs some changes prior its publications.

I am wondering why Authors use L. braziliensis antigen instead of L. infantum antigen or on the other hand do not infected the hamsters with L. braziliensis. Please provide a clear justification of it in the main text.

 The safety and toxicity evaluation of the nanoformulations should be also performed in-vitro. Did the author tested these? Please provide the results, if published elsewhere, please add the reference.

The authors should decide if call the not infected group as "control", "negative"  or "non-infected" I.E: Line 144 "The cut off was established by the average of the optical density (OD) of eight non-infected hamsters (preimmune group) plus two standard deviations measured in the negative control samples." 

The statistical significance reported in the figures are not clear, usually the significance is reported with asteriscs. Moreover the decision to use the letter "C" as a reference for the "Control" is confusing particularly when a image call "C" is present and no more detailes are given in the legend.

The most important statistical significances are between LBSAP and LBPSmP/G, I would suggest remarking these differences.

Figure 2, please present on the left the pictures that you describe first and on the right the pictures that you describe as second. 

Author Response

Revisor 1 comments:

“I am wondering why Authors use L. braziliensis antigen instead of L. infantum antigen or on the other hand do not infected the hamsters with L. braziliensis. Please provide a clear justification of it in the main text.”

We acknowledge the Reviwere#1 for this comment. We have included additional information in lines 412 to 419, as follows: “(…) In previous works was already demonstrated the cross protection of L. braziliensis crude antigens against L. infantum infection, besides the genetic homology between these parasite species [17–19]. Giunchetti et al. (2007) [17] demonstrated a significant increase in nitric oxide (NO) levels in dogs vaccinated with LBSap in an in vitro cell proliferation assay stimulated with soluble L. chagasi antigen. In this study was showed a protective response triggered by L. braziliensis plus saponin vaccine against visceral leishmaniasis. (…)”

“The safety and toxicity evaluation of the nanoformulations should be also performed in-vitro. Did the author tested these? Please provide the results, if published elsewhere, please add the reference”.

We have included the requested information (Lines 135 to 138): “(…) Previous studies from our group had already demonstrated that L. braziliensis crude antigen combined or not to saponin are safe and innocuous in in vitro, pre-clinic and clinic models and promising vaccines against CVL [17–19]. (…)”.

“The authors should decide if call the not infected group as "control", "negative"  or "non-infected" I.E: Line 144 "The cut off was established by the average of the optical density (OD) of eight non-infected hamsters (preimmune group) plus two standard deviations measured in the negative control samples." 

We have included additional information to clarify this issue (Lines 191 to 194), as follows: “(…) The cut off was established by the average of the optical density (OD) of eight non-infected hamsters (preimmune group related to randomized samples in hamster of all groups before the immunization protocol) plus two standard deviations measured for these samples. (…)”.

“The statistical significance reported in the figures are not clear, usually the significance is reported with asteriscs. Moreover the decision to use the letter "C" as a reference for the "Control" is confusing particularly when a image call "C" is present and no more detailes are given in the legend”.

We decided to use acronyms instead asterisks for show statistically significant data, as we have a great number of groups. In this way it would be easier to visualize the results. However, we improved the Figure descriptions as suggested; Please, see lines 280-281, 307-308, 341-342, 352-353, 371-372.

“The most important statistical significances are between LBSAP and LBPSmP/G, I would suggest remarking these differences.”

This difference was clear when compared LBSap, LBPSmG, and LBPSmG to C (control), SAP (saponin), and LB (L. braziliensis crude antigen) regarding parasite burden accessed through qPCR; we added further differences, and highlight in lines 363-367, as follows: “(…) Notably, the percentage of parasite load reduction in the LBSap, LBPSmG, and LBPSmP groups were 90%, 88%, and 94%, respectively, when compared to SAP group; 88%, 86%, and 93%, respectively, when compared to LB group, and 84%, 81%, and 90%, respectively, when compared to C (Control) group. (…)”.

“Figure 2, please present on the left the pictures that you describe first and on the right the pictures that you describe as second.”

Lines 294 to 296; and 323 to 325.

Reviewer 2 Report

The topic of this study is relevant to leishmaniasis management. Also, the results give a new perspective to developing vaccines. I detected some concerns. The comments are in the attached file.

Author Response

Revisor 2 comments:

Was animal location randomized? And how was it performed?

The sample size, and animals per group was determined according to sample calculation submitted to the institution ethical committee (CEUA-UFMG, protocol 385/2013) based on Moreira Nd, Vitoriano-Souza J, Roatt BM, Vieira PM, Ker HG, de Oliveira Cardoso JM, Giunchetti RC, Carneiro CM, de Lana M, Reis AB. Parasite burden in hamsters infected with two different strains of leishmania (Leishmania) infantum: "Leishman Donovan units" versus real-time PCR. PLoS One. 2012;7(10):e47907. doi: 10.1371/journal.pone.0047907. Epub 2012 Oct 24. PMID: 23112869; PMCID: PMC3480442. See line 113-114.

We have included the following sentence in the revised version of manuscript (lines 160 to 163): “(…) The animals enrolled in the study (n=72 hamster; n=8/group) were kept in ventilated racks with food and water ad libitum throughout the study, with intermittent light/dark cycles every 12 h, submitted to fully-blinded preclinical interventions.  (…)”.

To separate the number from the unit of measure. Review throughout the manuscript.

It was separated in the whole text.

Was the person that treated the animals blinded? Also, who measured the effect was the same that performed intervention?

The experiment was fully coded (fully-blinded preclinical interventions), from the initial identification of the animals to the final analyses, which ensured the unbiased treatment of data.

We have included additional information in lines 160-163: “(…) The animals enrolled in the study (n=72 hamster; n=8/group) were kept in ventilated racks with food and water ad libitum throughout the study, with intermittent light/dark cycles every 12 h, submitted to fully-blinded preclinical interventions. (…)”.

What is the vehicle? And volume?

We added the missing information in lines 171-172

How were the animals euthanized?

As requested, we have included euthanasia information (Lines 177-179), as follows: “(…) the animals were euthanized by the sodium pentobarbital overdose (150 mg / Kg), injected intraperitoneally. (…)”

Remove a period.

Line 183.

Vehicle used?

Line 187.

Vehicle?

Line 187.

How many animals were used for this test?

Lines 197 to 198.

How many animals per group were used?

Lines 160; 210 to 211.

What are the speed and time used?

Lines 213-214.

What is the sensibility of method?

For DNA extraction was used a MN kit that allows up to 90% of yield (was obtained up to 500 ng/mL). Also, DNA integrity was checked through electrophoresis and resolution in agarose gel.  The quality of extraction was accessed by quantification in the Nanodrop where was observed the ratio of purity of the samples.

Risk of bias of reporting data.

After euthanasia the organs (spleen and liver) from the animals included in the study (eight animals / eight groups) were weighed and photographed in a Petri dish over a graph paper; all the data was compiled. We did not consider necessary to include photographs and organ weight data in this publication. Indirectly, the veracity of the data can also be inferred by the frequency of lesions in the splenic and hepatic tissues showed in the histopathological analyses.

Where is statistical analysis? (Limiting dilution assay)

Please, see the lines 351 to 352 in the Fig. 4, as follows: “(…) Statistical differences (p <0.05, Mann-Whitney) between groups were indicated by acronyms in the correlated group (s). (…)”

Where is statistical analysis? (qPCR)

Please, see the lines 251 to 253, as follows: “(…) Considering the nonparametric nature of real time PCR data, the Kruskal-Wallis test was used, followed by Dunn’s test. (…)”.

Based on Melting curve?

The parasite burden was inferred through the difference between the amount of b-actin kDNA and the amount of the hamster’s sample kDNA. Log and Ct values was plotted in a linear curve to estimate the parasite per each 106 host cells; as included in lines 241-243, as follows: “(…) and the results were plotted using the standard curve method (parasite burden was inferred through the difference between the amount of b-actin kDNA and the amount of the hamster’s sample kDNA). (…)”.

This is a risk of bias of reporting data.

After euthanasia the organs (spleen and liver) from the animals included in the study (eight animals / eight groups) were weighed and photographed in a Petri dish over a graph paper; all the data was compiled. We did not consider necessary to include photographs and organ weight data in this publication. Indirectly, the veracity of the data can also be inferred by the frequency of lesions in the splenic and hepatic tissues showed in the histopathological analyses.

What are the study limitations?

Please, see lines 464 to 467, as follows: “(…) Although immunophenotyping assays were not performed, the parameters evaluated in the vaccinated animals suggest the occurrence of a protective cellular profile in this pre-clinical assay. (…)”.